# Salivary Oxytocin Concentration Changes during a Group Drumming Intervention for Maltreated School Children

**DOI:** 10.3390/brainsci7110152

**Published:** 2017-11-16

**Authors:** Teruko Yuhi, Hiroaki Kyuta, Hisa-aki Mori, Chihiro Murakami, Kazumi Furuhara, Mari Okuno, Masaki Takahashi, Daikei Fuji, Haruhiro Higashida

**Affiliations:** 1Department of Basic Research on Social Recognition, Research Center for Child Mental Development, Kanazawa University, Kanazawa 920-8640, Japan; furururukz.999@gmail.com (K.F.); brainsci@med.kanazawa-u.ac.jp (M.O.); haruhiro@med.kanazawa-u.ac.jp (H.H.); 2Lumbini Gakuen Ayabe, A Short-Term Therapeutic Institution for Emotionally Disturbed Children, Social Welfare Juridical Corporation Lumbini-en, Ayabe, Kyoto 629-1244, Japan; kyuuta.hiroaki@rouge.plala.or.jp (H.K.); hkty.08450888@gmail.com (H.M.); muracami814@yahoo.co.jp (C.M.); ize00157@nifty.com (M.T.); daikei.fuji@gmail.com (D.F.)

**Keywords:** child abuse, maltreatment, intervention, drum playing, salivary, oxytocin

## Abstract

Many emotionally-disturbed children who have been maltreated and are legally separated from their parents or primary caregivers live in group homes and receive compulsory education. Such institutions provide various special intervention programs. Taiko-ensou, a Japanese style of group drumming, is one such program because playing drums in a group may improve children’s emotional well-being. However, evidence for its efficacy has not been well established at the biological level. In this study, we measured salivary levels of oxytocin (OT), a neuropeptide associated with social memory and communication, in three conditions (recital, practice, and free sessions) in four classes of school-aged children. Following the sessions, OT concentrations showed changes in various degrees and directions (no change, increases, or decreases). The mean OT concentration changes after each session increased, ranging from 112% to 165%. Plasma OT concentrations were equally sensitive to drum playing in school-aged boys and girls. However, the difference between practice and free play sessions was only significant among elementary school boys aged 8–12 years. The results suggest that younger boys are most responsive to this type of educational music intervention.

## 1. Introduction

Childhood maltreatment represents the most potent predictor of poor mental health across the lifespan [1,2,3]. Such adversity increases the risk of a wide range of psychiatric disorders, including reactive attachment disorder (RAD) and autism spectrum disorder (ASD) [4,5,6,7]. Children with ASD and RAD experience similar difficulties with social relationships [8,9,10], but there appears to be a difference in the quality of their social interactions. In most cases it is possible to differentiate between children with ASD and children with RAD via structured observation. The most important difference is that RAD is associated with neglect or maltreatment, whereas ASD usually is not [8,9].

Children who are found to have been maltreated are entrusted to a foster parent or admitted to a children’s home, a short-term therapeutic institution for emotionally disturbed children, or a children’s self-reliance support facility [11,12,13]. In Japan, emotionally-disturbed children, including those who have been maltreated, are either admitted to a short-term therapeutic institution or treated as out-patients, and receive consultation and other assistance. Children who are resident in such institutions learn to adapt to social life while living together in groups, and also receive compulsory education [12,13].

It is increasingly recognized that music-making interventions can enhance mental health [14,15,16]. Among them, drumming has long been a part of traditional healing rituals worldwide, and is increasingly used as a therapeutic strategy [17,18,19]. The features of group drumming programs are known to facilitate mental health recovery. The findings of previous studies support the concept of “creative practice as mutual recovery”, demonstrating that group drumming provides a creative and mutual learning space in which mental health recovery can take place [17,18,20]. These reports indicate a shift away from a pro-inflammatory toward an anti-inflammatory immune profile. Consequently, the psychological benefits of group drumming and the underlying biological effects support its therapeutic potential for mental health [18,20]. Drumming is a complex composite intervention with the potential to modulate specific neuroendocrine and neuroimmune parameters in the opposite direction to that of the classic stress response [20].

Recent studies have suggested that oxytocin (OT) plays a role in social memory and behavior [21,22,23,24,25,26,27,28]. OT has positive effects on social and emotional processes in healthy subjects and some individuals diagnosed with a variety of psychiatric disorders [29,30,31,32,33]. Nasal application of OT in subjects with ASD with or without comorbid intellectual disability has been shown to improve social interactions [34,35,36,37,38].

This study was conducted in a facility where children with mild emotional disturbance live and study together. Most of them are isolated from their parents and primary caregivers, owing to maltreatment and neglect. The institute offers special educational intervention programs, such as group drumming, to improve their emotional well-being. However, the facility does not use any biological measures to monitor the beneficial effects of group drumming. Salivary OT can be measured in human saliva, suggesting that it may be a reliable biomarker [39,40,41,42,43,44,45,46]. Therefore, we examined whether maltreated school-aged children exhibited changes in OT concentrations during educational sessions based on Japanese Taiko group drumming (Appendix A) [47]. In this study, we measured salivary OT concentrations in children before and after recital or practice drumming sessions and compared them with those in free play sessions. We also analyzed changes in OT concentrations by classifying participants by sex and age, such as boys in the elementary school (8–12 years old) and junior high school (13–15 years old), or girls in the elementary school. We also measured OT concentrations in the children’s drumming instructors.

## 2. Materials and Methods

### 2.1. Participants

The study recruited 23 boys and five girls aged 8–15 years as voluntary participants (Table 1). The children had lived for between 0.5 and six years in the short-term therapeutic institution for emotionally-disturbed children, run by the Social Welfare Juridical Corporation Lunbini-en (Ayabe, Kyoto, Japan). They attended either the elementary or the junior high school attached to the Lunbini-en. The children were placed in this facility by municipal or prefectural child guidance centers because they were considered to be in need of daily life guidance due to their family environment. We obtained data from three male drumming instructors who taught at the schools (36.2 ± 3 years old).

### 2.2. Ethics Statement

The study was approved as a non-invasive medical study by the institutional review board of the Social Welfare Juridical Corporation Lunbini-en in 2014 and by Kanazawa University Graduate School of Medicine in 2015 (approval number #2012-1). The study was performed according to the Declaration of Helsinki and the Ethical Guidelines for Clinical Studies of the Ministry of Health, Labor and Welfare of Japan. After they had been given a complete explanation of the study, all of the participants and their caregivers or the child welfare officers in the child guidance centers provided written informed consent. The participants were told that they could choose not to supply their saliva on each occasion, even after agreeing to participate in the study.

### 2.3. Assessment 

The children’s salivary OT levels were assessed during 19 sessions of group Taiko drumming (Appendix A) from July 2015 to December 2016. The children played freely for the first 10 min. Saliva was collected in a sterile 15-mL polyproprylene tube (Greiner Bio-one Co. Ltd., Tokyo, Japan). Two to five minutes after rinsing with water, the children’s mouths filled with newly-secreted saliva. They bit the tube in their mouths and secreted saliva directly into the tube by chewing for 2–4 min. This method was less stressful for such children than using a cotton swab and they were able to complete it by themselves without teachers’ assistance. Then they participated in a 5–60 min (14.1 ± 3.5 min, *n* = 14) during recitals on stages in front of an audience (Appendix A), or six 80–155 min (108 ± 10.2 min, *n* = 6) practice sessions in a hall at the Lunbini-en (Appendix A, upper panels). Saliva was collected a second time after 10 min of all sessions. As a control, saliva was collected 10 min before and after the free play sessions (during which they usually moved, chatted, and read freely) for 90–200 min (118 ± 16.5, *n* = 6) in the same hall or playground of the Lunbini-en (Appendix A, lower panels).

### 2.4. Saliva Collection and Analysis 

The saliva samples (0.3–0.8 mL) were collected in polyproprylene tubes and were immediately frozen in dry ice and stored at −20 °C, as described previously [48]. Three days later, they were thawed and centrifuged twice at 4 °C at 1500× *g* for 15 min. The samples were divided into 1.5-mL microtubes, each containing 100 µL, and kept again at −80 °C until assay.

Salivary OT was measured using a 96-plate commercial OT-ELISA kit (Enzo Life Sciences, Farmingdale, NY, USA), as described previously [48,49]. Measurements were performed in duplicate. Samples (100 µL) without fractionation were treated according to the manufacturer’s instructions. The optical density of the samples and standards was measured at wavelengths of 405 and 590 nm by a microplate reader (Bio-Rad, Richmond, CA, USA). Sample concentrations were calculated by MatLab-7 (MathWorks, Inc., Natick, MA, USA) according to the relevant standard curve.

### 2.5. Statistical Analysis

Two-tailed Student’s *t* tests were used for single comparisons between two groups. One- or two-way analyses of variance were used for data with two or three components, respectively. Post hoc comparisons were performed only when the main effect was statistically significant. The *p*-values of the multiple comparisons were adjusted using Bonferroni’s correction. All data from in vivo and in vitro studies are shown as means ± s.e.m. In all analyses, *p* < 0.05 was taken to indicate statistical significance. All of the analyses were performed using STATA data analysis and statistical software (Stata Corp. LP, College Station, TX, USA).

## 3. Results

Table 1 shows the demographic characteristics of the participants in the traditional Japanese Taiko group. A total of 23 boys and five girls from the elementary and junior high schools were involved in the sessions, during which three male teachers instructed them to play a piece of music.

The OT concentrations in the saliva collected from the children and adults before the performance and control sessions were determined and used as the baseline OT levels (Table 2). The average baseline salivary OT level did not differ significantly between assessments for any of the groups, but the highest value was obtained from the elementary school boys. The OT concentrations in the saliva after different activities (recital, practice (lesson), and free) were plotted separately for the elementary school boys (Figure 1), junior high school boys (Figure 2), elementary school girls (Figure 3), and teachers (Figure 4). No significant differences were observed before and after sessions in the four groups of participants, except for teachers at the recitals (*p* < 0.01).

The difference of salivary OT concentrations before and after the activity was calculated. Since the conditions differed in terms of whether an instrument was played and the duration, we compared the changes in OT between non-obligatory activities and practice with the instrument with a similar time gap (Figure 5). Similarly, the difference was compared between recital and practice, both of which involved playing the drums (Figure 6). Significant changes in the ratio of salivary OT levels before and after activities and between playing practice and free sessions were observed only in the elementary school boys (two-way Student’s *t*-test, *n* = 11–34, *p* < 0.02).

## 4. Discussion

We examined the biological effects of playing music in a Japanese Taiko drumming group. This activity was performed less than once a week as part of an educational intervention for children living and studying in a short-term therapeutic institution for emotionally disturbed children, most of whom were legally separated from their parents or caregivers because of maltreatment [12,13]. The results indicated that the children’s mean salivary OT concentrations were increased to various degrees after the activity sessions. The smallest increase was observed in the elementary school boys during the practice session and the largest change was in the girls of the elementary school after the free session. However, the only significant difference was between practice and free sessions for boys aged 8–12 years (Figure 5), although a similar tendency was observed in the junior high school boys and elementary school girls, it was not significant. The change in OT concentration was in the reverse direction, i.e., higher during practice than free time, although in the teachers’ case it was during drumming instruction.

The reason such significant changes were observed only in younger boys may be that drum playing creates a better atmosphere for younger boys than older ones. In other words, drum playing may not be stressful for older boys but for younger boys because high school boys are already familiar with drumming. Drumming on a stage in front of an audience (recital) and playing for practice (rehearsal) may be essentially different for the players. In addition, the performing time was also different: recitals lasted only 14 min on average, compared with 108 min for practice. This makes it difficult to compare the three groups on the different conditions, but as the experiment did not interfere with the institutional routine, optimal ecological validity was maintained.

However, even shorter activities seemed to be stressful. The average OT concentrations were lower after group drumming recital and practice sessions than before. The effects were evident in boys, and seemed to be due to the group drumming rather than playing individually, which created some stress. The OT concentration changes before and after free play, as a control condition, were in the opposite direction.

In relation to our study, there are several interesting reports on playing music as a group, which describe differences between singing alone (solo) and in a group. Schladt et al. [50] reported that OT increased in the case of solo singing but not in choir. This suggests that, although singing in a group seems to be a joyful experience and one that facilitates bonding [51], it can be rather stressful. Singing lessons for professional or amateur singers are not the same, because professional singers are achievement-oriented, but amateurs sing for self-actualization or self-expression [52]. For the school children in the current study, it is likely that OT increased owing to the release of emotional tension in the recital or practice sessions. In the teachers, OT significantly increased in the recital sessions (Figure 4), probably because they were satisfied by the children’s successful performance.

Overall behavior while staying in this therapeutic institution generally improved: the children thought that the Lumbini-en is safe and secure, solved vigilance, and opened their hearts to the surroundings. They became to sleep well, to feel less anxiety, and less likely to panic with a trivial chance. They became to control themselves according to their opponents and situations. They participated in group activities and continued to go to the inside school. They tried to decide their own lives by themselves, grew self-confidence, and finally got to have the hope of going to alive. Such outcomes of individuals were assessed using the school’s behavior and study records. Child welfare officers and teachers rated the children’s behavior using a questionnaire. In addition, the officers and teachers felt that drumming can be a positive and effective educational intervention to improve children’s behavior.

Next, we considered the relationship between OT concentrations and individual’s behaviors. Participants were classified according to the initial OT levels and changes in OT levels before and after a single session. The drumming teachers reported that individuals in the group with OT levels in the recital and free sessions that were higher than average are restless, weaker to stimulation, autistic and hyperactive, compared with other individuals (Appendix A). In the teacher’s records, participants who showed higher rates of change in OT levels before and after recital or free sessions are relatively hyperactive compared with individuals in other classes (Appendix A). These results, interestingly, indicate that OT levels modified to a greater extent in a category of hyperactive boys and girls.

It is worth considering the well-known correlation between OT and group cohesion. It is interesting that in boys free play tended to lead to increased OT levels than did practical lessons. This may be due to the cohesiveness effect. To perform in a recital is stressful, particularly for children with disturbed social function, and is likely to reduce vagal activity and cohesiveness. Increases in OT concentrations correspond to increased vagal activity and vice versa, resulting from the brain mechanism as described in prairie voles [53]. In line with this, a study by Harmat and Theorell [54] showed dramatically reduced vagal activity during a concert compared with during a rehearsal, and particularly among those musicians who reported nervousness in recital.

This study had some limitations. Although the measures of salivary OT were taken two years apart, the number of school children involved was small. Opportunity sampling meant that the number of participants was not constant on each occasion.

It has been reported that salivary OT is frequently measured as the biomarker [41,43,55,56,57,58], however, though some of the saliva components interact with the labeled OT in the assay mixture [59]. To test this, we first measured samples spiked with 50, 100, 200, and 400 ρg/mL. Even the very high level was observed with no addition of spiked OT, suggesting an interaction with nonspecific antibody-interacting substances. Even with interacting substances in the samples, the EIA monitored concentrations were proportional to the spiked OT concentrations, suggesting that monitored values are useful for calculating the ratio between values. Therefore, the difference of OT concentrations before and after the sessions shown in Figure 5 and Figure 6 seems to be reliable.

This study did not address behavioral changes in the children at school. Such tests should be performed using a standard questionnaire survey and the children’s school records. Teachers in the Institution have the impression that drumming improves the behavior of children living in care. In general, they report that the caregiver-teacher-child relationship improves within the first month up to the end of the year.

In conclusion, this study found that maltreated children responded reasonably well and benefitted from a group drumming intervention, as indicated by their increased salivary OT concentrations. The intervention was effective because the children experienced stress, and the stress-relax cycle is linked to the capacity to control emotional reaction.

## Figures and Tables

**Figure 1 brainsci-07-00152-f001:**
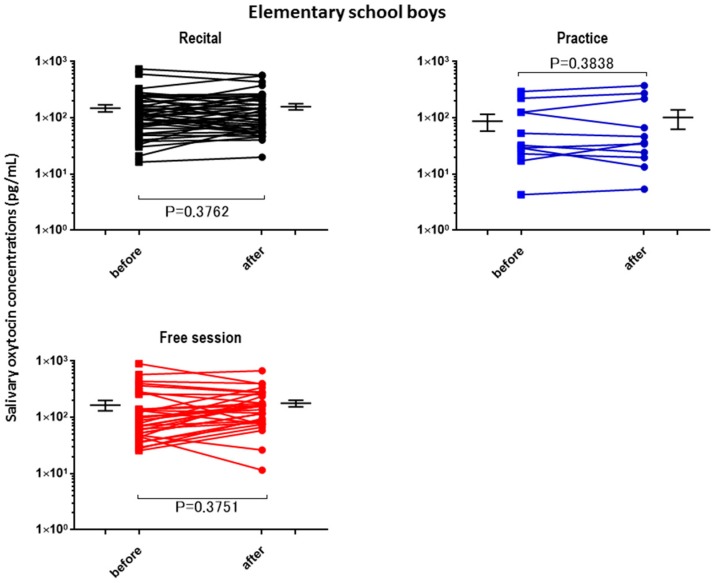
Changes in elementary school boys’ oxytocin levels before and after a single play session. Oxytocin levels are for the first (before) and second (after) salivary samples. Saliva was collected at each recital (*n* = 52), practice (*n* = 11), and free play (*n* = 33) session. *p* values are for two-tailed Student’s *t*-tests.

**Figure 2 brainsci-07-00152-f002:**
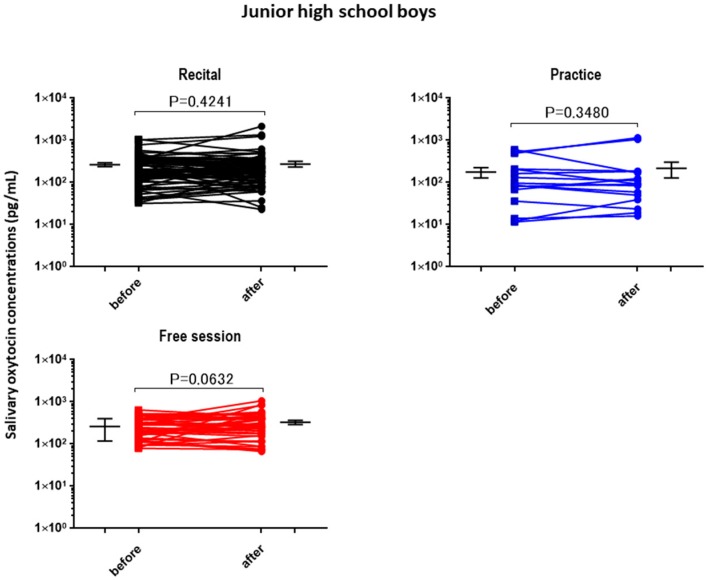
Changes in junior high school boys’ oxytocin levels before and after a single play session. Oxytocin levels are for the first (before) and second (after) salivary samples. Saliva was collected at each recital (*n* = 61), practice (*n* = 17), and free play (*n* = 55) session. *p* values are for two-tailed Student’s *t*-tests.

**Figure 3 brainsci-07-00152-f003:**
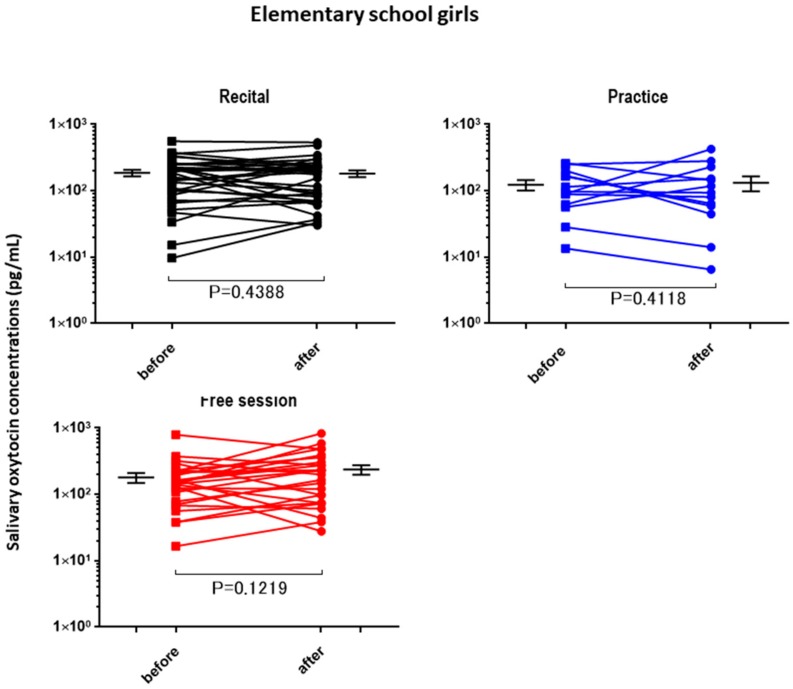
Changes in elementary school girls’ oxytocin levels before and after a single play session. Oxytocin levels are for the first (before) and second (after) salivary samples. Saliva was collected at each recital (*n* = 33), practice (*n* = 14), and free play (*n* = 30) session. *p* values are for two-tailed Student’s *t*-tests. A significant difference was found for the recital session.

**Figure 4 brainsci-07-00152-f004:**
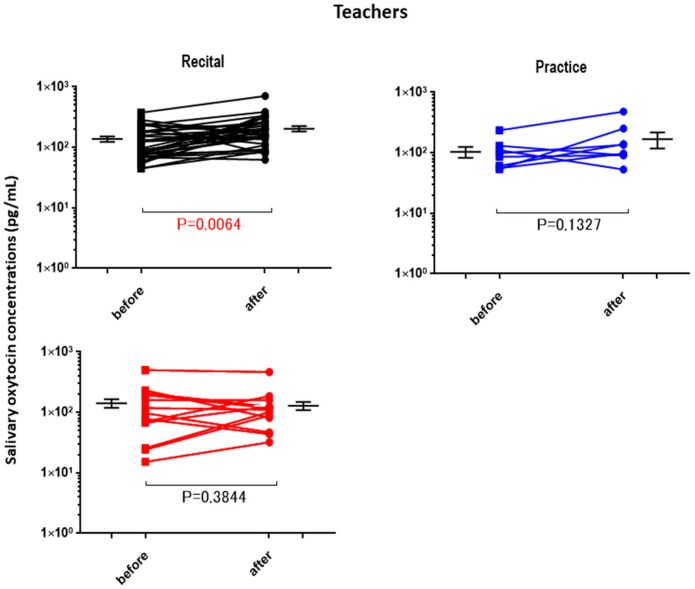
Changes in teachers’ oxytocin levels before and after a single play session. Oxytocin levels are for the first (before) and second (after) salivary samples. Saliva was collected at each recital (*n* = 33), practice (*n* = 8), and free play (*n* = 14) session. *p* values are for two-tailed Student’s *t*-tests.

**Figure 5 brainsci-07-00152-f005:**
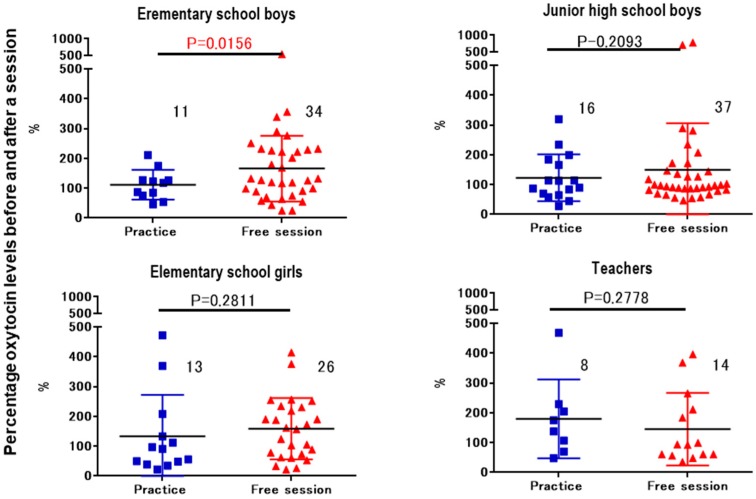
Changes in oxytocin levels before and after a single session. Percentage of oxytocin levels for the first (before) saliva over the second (after) saliva samples. Saliva was collected at practice and free play sessions from elementary school boys, junior high school boys, elementary school girls, and teachers. *p* values are for two-tailed Student’s *t*-tests.

**Figure 6 brainsci-07-00152-f006:**
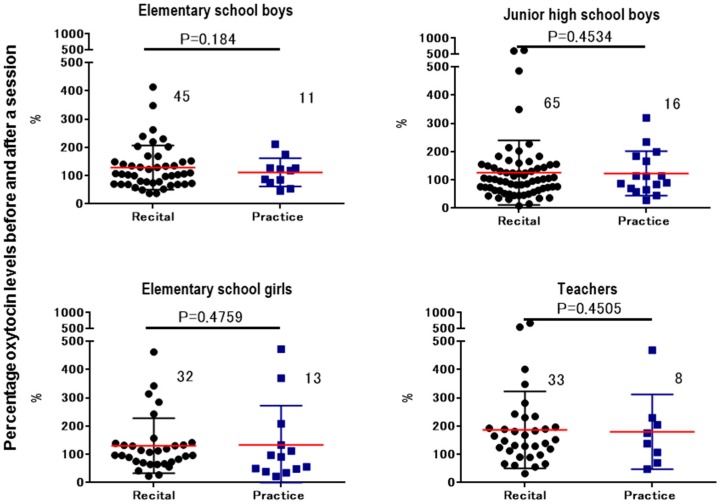
Changes in oxytocin levels before and after a single session. Percentage of oxytocin levels for the first (before) saliva over the second (after) saliva samples. Saliva was collected at recital and practice sessions from elementary school boys, junior high school boys, elementary school girls, and teachers. *p* values are for two-tailed Student’s *t*-tests.

**Table 1 brainsci-07-00152-t001:** Demographic data.

Standing	Elementary School	Junior High School	Teacher
Age (years)	8–12 (9.9 ± 0.9)	13–15 (14.4 ± 0.4)	33–40 (36.2 ± 3)
Gender			
male	9	13	3
female	5	0	0
Intervention duration (min)			
None	118 ± 16.5 (*n* = 6)		
Practice	108 ± 10.2 (*n* = 5)		
Recital	14.1 ± 3.5 (*n* = 12)		

**Table 2 brainsci-07-00152-t002:** Baseline concentrations of oxytocin (pg/mL).

	Boys		Girls	Adults	
	Elementary School	Junior High School	Elementary School	Teachers	
Free activity	166 ± 34 (32)	259 ± 23 (39)	179 ± 30 (26)	142 ± 34 (14)	*F*_3,109_ = 3.36 (*p* = 0.0215)
Practice	83 ± 29 (14)	176 ± 48 (16)	123 ± 22 (13)	103 ± 21 (8)	*F*_3,45_ = 1.61 (*p* = 0.1992)
Recital	149 ± 21 (44)	265 ± 31* ^#^ (63)	186 ± 20 (34)	138 ± 14 (33)	*F*_3,173_ = 6.09 (*p* = 0.0006)
	*F*_2,84_ = 1.01 (*p* = 0.3681)	*F*_2,122_ = 1.76 (*p* = 0.1768)	*F*_2,70_ = 1.23 (*p* = 0.2989)	*F*_2,58_ = 0.01 (*p* = 0.9884)	

One-way ANOVA analysis of each matrics is shown. * ^#^
*p* < 0.01 for recital in elementary school boys and teachers (Bonferroni’s test).

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
