# Peer review of "Salivary Oxytocin Concentration Changes during a Group Drumming Intervention for Maltreated School Children"

_brainsci, 2017, doi:10.3390/brainsci7110152_

Round 1

Reviewer 1 Report

Yuhi et al measured OT in saliva in emotionally disturbed school boys and girls in response to drumming exercises (Taiko), a Japanese style of group drumming. Samples were taken under three conditions : recital, practice, and free sessions in four classes of school-aged children. This is an interesting therapeutical approach, however, there are several points which need to be considered.

1.      In the introduction they state that “Salivary OT can be measured and intranasal OT administration is evident in human saliva, suggesting that it may be a reliable biomarker ..”. This essentially needs to be corrected: Indeed endogenous OT can be measured reliably in saliva (see deJong T et al 2016, PNEC), but the context of intranasal OT should be strictly avoided, as all intranasal OT will appear to large extent in saliva simply by swallowing it. So, saliva OT might indeed be a biomarker, but NOT after intranasal application  of synthetic OT.

2.      A recent study (Frontiers in Human Sciences) by Schladt M et al 2017 demonstrated that another music experience, i.e. solo singing, but not choir singing, increases OXT in saliva, this should be mentioned and discussed, also in the context of single or group performance.

3.      It is stated that children rinsed their mouths with water prior to sampling? This will significantly dilute OT in saliva and may result in very varying levels of OT. Please explain and discuss in context of published methods

4.      The  figures would strongly profit from showing in addition to individual data also group means and the number of n in each group. Also, Figures could be meaningful summarized and combined to avoid too many simple figures.

5.      In fig 6 there is no “before” and “after” (legend).

6.      The scales of all figures should be altered to emphasize between values 100 and 500 pg/ml

7.      The conclusion that maltreated children responded reasonably well and benefitted from a group drumming intervention, as indicated by their increased salivary OT concentrations is a bit overdone, as there is little or even no increase in OT described in the figures, pls be a bit more specific in which children such an increase might be found.

8.      The last sentence of discussion is incomplete: The intervention was effective because the children experienced stress, and the stress-relax cycle is linked to the capacity to control

Author Response

Answers for comments from the Reviewer 1

In the introduction they state that “Salivary OT can be measured and intranasal OT administration is evident in human saliva, suggesting that it may be a reliable biomarker ..”. This essentially needs to be corrected: Indeed endogenous OT can be measured reliably in saliva (see deJong T et al 2016, PNEC), but the context of intranasal OT should be strictly avoided, as all intranasal OT will appear to large extent in saliva simply by swallowing it. So, saliva OT might indeed be a biomarker, but NOT after intranasal application  of synthetic OT.

We appreciated the cool comment and we agree with this comment. We deleted ‘ and intranasal OT administration is evident’ as follow (Lines 66-67): However, the facility does not use any biological measures to monitor the beneficial effects of group drumming. Salivary OT can be measured in human saliva, suggesting that it may be a reliable biomarker [39-46]. Therefore, we examined whether maltreated school-aged children exhibited changes in OT concentrations during educational sessions based on Japanese Taiko group drumming (Supplementary Figure 1) [47]. 

And as suggested, we added Jong’s paper as a reference 46

A recent study (Frontiers in Human Sciences) by Schladt M et al 2017 demonstrated that another music experience, i.e. solo singing, but not choir singing, increases OXT in saliva, this should be mentioned and discussed, also in the context of single or group performance.

Thank you very much for the latest information in oxytocin and music, especially in singing. The comment is the same as the reviewer 2’s first comment. According to the suggestion, we discussed as follow (Lines 203-211).: In relation to our study, there are several interesting reports on playing music as a group, which describe differences between singing alone (solo) and in a group. Schladt et al. [50] reported that OT increased in the case of solo singing but not in choir. This suggests that, although singing in a group seems to be a joyful experience and one that facilitates bonding [51], it can be rather stressful. Singing lessons for professional or amateur singers are not the same, because professional singers are achievement-oriented, but amateurs sing for self-actualization or self-expression [52]. For the school children in the current study, it is likely that OT increased owing to the release of emotional tension in the recital or practice sessions. In the teachers, OT significantly increased in the recital sessions (Fig. 4), probably because they were satisfied by the children’s successful performance.

And newly added 3 references suggested as Ref. #49, 50, 51.

3.      It is stated that children rinsed their mouths with water prior to sampling? This will significantly dilute OT in saliva and may result in very varying levels of OT. Please explain and discuss in context of published methods

In practically, after washing there are 3-5 minutes before collection. Thus, the mouth seemed to be filled with newly secreted saliva. Mended as follow (Lines 98-101): Two to five minutes after rinsing with water, the children’s mouths filled with newly secreted saliva. They bit the tube in their mouths and secreted saliva directly into the tube by chewing for 2-4 minutes. This method was less stressful for such children than using a cotton swab and they were able to complete it by themselves without teachers’ assistance

4.      The  figures would strongly profit from showing in addition to individual data also group means and the number of n in each group. Also, Figures could be meaningful summarized and combined to avoid too many simple figures.

Thank you very much for your scientific suggestions. We described ‘number’ in figures or figure legends. We also added means ± standard error, besides line graphs in Figs.1-4 and red lines among dots in Figs. 5-6. Because graphs became slightly complex, we did not combine them into integrated figures.

5.      In fig 6 there is no “before” and “after” (legend).

Thank you for your comment. I failed to add y-axis unit of measurement. Figs 5 and 6 the percentage of concentration (after /before x 100). To make clear, we indicated ‘%’ for the ordinate.

6.      The scales of all figures should be altered to emphasize between values 100 and 500 pg/ml

According to this suggestion, the scale of figures were modified, and outrange values were shown appropriately by splitting the scale. 

7.      The conclusion that maltreated children responded reasonably well and benefitted from a group drumming intervention, as indicated by their increased salivary OT concentrations is a bit overdone, as there is little or even no increase in OT described in the figures, pls be a bit more specific in which children such an increase might be found.

We appreciated the thoughtful suggestion. We carefully inspected behavioral records of each participant during the current study. Discussing with teachers, we found some tendency. Because these documents are not a kind of results verified by a statistical analysis, we thought it may be better to describe in Discussion and show in new supplementary figures 1 and 2, rather than a part of results(Lines 216-225).: Next, we considered the relationship between OT concentrations and individual’s behaviors. Participants were classified according to the initial OT levels and changes in OT levels before and after a single session. The drumming teachers reported that individuals in the group with OT levels in the recital and free sessions that were higher than average are restless, weaker to stimulation, autistic and hyperactive, compared with other individuals (Supplementary Figure 4). In the teacher’s records, participants who showed higher rates of change in OT levels before and after recital or free sessions are relatively hyperactive compared with individuals in other classes (Supplementary Figure 5). These results, interestingly, indicate that OT levels modified to a greater extent in a category of hyperactive boys and girls.

8.      The last sentence of discussion is incomplete: The intervention was effective because the children experienced stress, and the stress-relax cycle is linked to the capacity to control

Thank you very much. This is our simple mistake. We added two words (emotional reaction) at line 252.

Reviewer 2 Report

The study has potential but I have several suggestions for improvement before final decision about publishability.

First of all, it is difficult to find relevant literature in this trans-disciplinary field. But there are also studies of individual singing and choir singing that show plasma oxytocin increases during singing which ought to be quoted.

Grape, C., Sandgren, M., Hansson, L-O., Ericson, M. and Theorell, T. (2003). Does singing promote well-being? An empirical study of professional and amateur singers during a singing lesson. Integrative Physiological and Behavioral Science. 38, 65-74

Kreutz G Does singing facilitate social bonding? Music and Medicine 2014 6:51-60

Secondly a general remark is that oxytocin levels are highly skewed which is also illustrated clearly in the ratio diagrams which show that the percentage change from before to after session has a skewed distribution. Therefore a common strategy in oxytocin studies is to transform the concentrations logarithmically and perform all the statistical computations with transformed data and present back-transformed means with 95% confidence intervals (which is geometric means with confidence limits). This may also affect the significance levels since extreme concentrations (which are likely to be true) do not weigh so heavily in the standard deviations after such a transformation.

Thirdly the authors count each session in their n:s as a separate “individual”. This may perhaps be warranted but needs to be discussed. After all, individual values are auto-correlated and it is difficult for me to

There are places where the language needs improvement. For instance, already in the abstract the authors state:

….the mean  OT concentration changes after each session increased from 112% before the sessions to 165% after….

This needs to be explained

and

…Salivary OT is frequently measured as a biomarker [48-53]. During measurement by the enzyme immunoassay, saliva was not extracted in the current experiment, as there are some reports measured without extraction [39,47]. Since the volume of saliva collected was not constant, but varied from 300 μL to 800 μL, which was insufficient to conduct duplicate measurements…..

It did not help me to go to line 113 in the methods section, it is still not clear what this means

I think the authors have missed the opportunity to make an interesting discussion regarding the well-known correlation between oxytocin and group cohesion in this work. It is interesting that free play elicits more oxytocin increase than practice in boys (with logarithmic transformation the p-value for the junior high school boys might improve). This may be due to the cohesiveness effect. To perform in a recital is stressful (maybe partiularly for children with disturbed social function) and is likely to reduce vagal activity and cohesiveness. Increase in oxytocin concentration corresponds to increased vagal activity and vice versa. In line with this a study has shown a dramatically reduced vagal activity during concert compared to rehearsal and particularly among those musicians who reported nervousness in recital.

Harmat, L. and Theorell, T. Heart rate variability during singing and flute playing. Music and Medicine. 2(1), 2010

Author Response

Answers for comments from the Reviewer 2

The study has potential but I have several suggestions for improvement before final decision about publishability.

First of all, it is difficult to find relevant literature in this trans-disciplinary field. But there are also studies of individual singing and choir singing that show plasma oxytocin increases during singing which ought to be quoted. Grape, C., Sandgren, M., Hansson, L-O., Ericson, M. and Theorell, T. (2003). Does singing promote well-being? An empirical study of professional and amateur singers during a singing lesson. Integrative Physiological and Behavioral Science. 38, 65-74. Kreutz G Does singing facilitate social bonding? Music and Medicine 2014 6:51-60

We appreciated for introducing a few reference which related to solo or choir singing. Singing is another important area in music intervention, as a comparable activity of drumming. We discussed these points as follow (Lines 206-215): In relation to our study, there are several interesting reports on playing music as a group, which describe differences between singing alone (solo) and in a group. Schladt et al. [50] reported that OT increased in the case of solo singing but not in choir. This suggests that, although singing in a group seems to be a joyful experience and one that facilitates bonding [51], it can be rather stressful. Singing lessons for professional or amateur singers are not the same, because professional singers are achievement-oriented, but amateurs sing for self-actualization or self-expression [52]. For the school children in the current study, it is likely that OT increased owing to the release of emotional tension in the recital or practice sessions. In the teachers, OT significantly increased in the recital sessions (Fig. 4), probably because they were satisfied by the children’s successful performance.

Secondly a general remark is that oxytocin levels are highly skewed which is also illustrated clearly in the ratio diagrams which show that the percentage change from before to after session has a skewed distribution. Therefore a common strategy in oxytocin studies is to transform the concentrations logarithmically and perform all the statistical computations with transformed data and present back-transformed means with 95% confidence intervals (which is geometric means with confidence limits). This may also affect the significance levels since extreme concentrations (which are likely to be true) do not weigh so heavily in the standard deviations after such a transformation.

We appreciated your sincere comments. We agree with the point that the clustered lines are not so meaningful. We transformed the ordinate of new Figs. 1-4 to the logarithmic scale.  

Thirdly the authors count each session in their n:s as a separate “individual”. This may perhaps be warranted but needs to be discussed. After all, individual values are auto-correlated and it is difficult for me to

We appreciated insightful comment. We think that this comment is essentially same as the comment of the reviewer 1. According to this suggestion, we payed attention to individuals. We classified these boys and girls according to their OT levels shown in new Supplementary Figures 4 and 5. We described characteristics of these individuals, especially for responsible persons.

There are places where the language needs improvement. For instance, already in the abstract the authors state:

.the mean  OT concentration changes after each session increased from 112% before the sessions to 165% after….

This needs to be explained

Thank you for this point. We mended as follow (Line 23): ranging from 112% to 165%

and

Salivary OT is frequently measured as a biomarker [48-53]. During measurement by the enzyme immunoassay, saliva was not extracted in the current experiment, as there are some reports measured without extraction [39,47]. Since the volume of saliva collected was not constant, but varied from 300 μL to 800 μL, which was insufficient to conduct duplicate measurements…..

It did not help me to go to line 113 in the methods section, it is still not clear what this means

Thank you very much. According to this suggestion, we deleted these description in the old version (3 sentences lines from 210-213 in old Discussion). Instead, we cited a very recent article which describe the validity to the current OT EIA kit method,(Ref. 49): MacLean EL, Gesquiere LR, Gee N, Levy K, Martin WL, Carter CS. Validation of salivary oxytocin and vasopressin as biomarkers in domestic dogs. J Neurosci Methods. 2017, 293, 67-76.

 .

I think the authors have missed the opportunity to make an interesting discussion regarding the well-known correlation between oxytocin and group cohesion in this work. It is interesting that free play elicits more OT increase than practice in boys (with logarithmic transformation the p-value for the junior high school boys might improve). This may be due to the cohesiveness effect. To perform in a recital is stressful (maybe partiularly for children with disturbed social function) and is likely to reduce vagal activity and cohesiveness. Increase in OT concentration corresponds to increased vagal activity and vice versa. In line with this a study has shown a dramatically reduced vagal activity during concert compared to rehearsal and particularly among those musicians who reported nervousness in recital. Harmat, L. and Theorell, T. Heart rate variability during singing and flute playing. Music and Medicine. 2(1), 2010.

We appreciated this scientific comment with a recommended reference. We discussed about oxytocin, stress and vagal tone in relation to our result along with your suggestion, as follow (Line 214-220): It is worth considering the well-known correlation between OT and group cohesion. It is interesting that in boys free play tended to lead to increased OT levels than did practical lessons. This may be due to the cohesiveness effect. To perform in a recital is stressful, particularly for children with disturbed social function, and is likely to reduce vagal activity and cohesiveness. Increases in OT concentrations correspond to increased vagal activity and vice versa, resulting from the brain mechanism as described in prairie voles [53]. In line with this, a study by Harmat and Theorell [54] showed dramatically reduced vagal activity during a concert compared with during a rehearsal and particularly among those musicians who reported nervousness in recital.

Round 2

Reviewer 1 Report

In the revised version of the manuscript entitled "

Salivary Oxytocin Concentration Changes during a Group Drumming Intervention for Maltreated School Children" the authors have considered all the critical points. There are only a few comments to be addressed:                

1. Please specifiy the behaviour: “Overall behavior while staying in this therapeutic institution generally improves”          

2. Some sentence throughout the ms need correction such as “The children usually recover and left along individual conditions from it.”(discussion).          

Author Response

(Reviewer 1)

Salivary Oxytocin Concentration Changes during a Group Drumming Intervention for Maltreated School Children" the authors have considered all the critical points. There are only a few comments to be addressed:                

1. Please specifiy the behaviour: “Overall behavior while staying in this therapeutic institution generally improves”         

Thank you for your comment. We described for detail in the text (line 212-218).

2. Some sentence throughout the ms need correction such as “The children usually recover and left along individual conditions from it.”(discussion).     

  Thank you for your suggestion. The sentence was deleted.

Reviewer 2 Report

You have made changes appropriately

Author Response

apriciate